# Effects of miR-103 by negatively regulating *SATB2* on proliferation and osteogenic differentiation of human bone marrow mesenchymal stem cells

**Hao Lv, Huashan Yang, Yuanrui Wang**₀*

Department of Trauma Center, Jinan Central Hospital Affiliated to Shandong First Medical University, Jinan, Shandong Province, P.R. China

* wangyrjinan@sina.com

## Abstract

### Background

The proliferation and osteogenic differentiation of human bone marrow mesenchymal stem cells (HBMScs) are modulated by a variety of microRNAs (miRNAs). SATB homeobox 2 (*SATB2*) is a critical transcription factor that contributes to maintain the balance of bone metabolism. However, it remains unclear how the regulatory relationship between miR-103 and *SATB2* on HBMScs proliferation and osteogenic differentiation.

### Methods

HBMScs were obtained from Cyagen Biosciences and successful induced osteogenic differentiation. The proliferation abilities of HBMScs after treatment with agomiR-103 and antagomiR-103 were assessed using a cell counting Kit-8 (CCK-8) assay, and osteogenic differentiation was determined using alizarin red S staining and alkaline phosphatase (*ALP*) activity assay. The expression levels of miR-103, *SATB2*, and associated osteogenic differentiation biomarkers, including RUNX family transcription factor 2 (*RUNX2*), bone gamma-carboxyglutamate protein (*BGLAP*), and secreted phosphoprotein 1 (*SPP1*), were evaluated using real-time qPCR and Western blot. The regulatory sites of miR-103 on SATB2 were predicted using bioinformatics software and validated using a dual luciferase reporter assay. The underlying mechanism of miR-103 on *SATB2*-mediacted HBMScs proliferation and osteogenic differentiation were confirmed by co-transfection of antagomiR-103 and *SATB2* siRNA.

### Results

The expression of miR-103 in HBMScs after induction of osteogenic differentiation was reduced in a time-dependent way. Overexpression of miR-103 by transfection of agomiR-103 suppressed HBMScs proliferation and osteogenic differentiation, while silencing of miR-103 by antagomiR-103 abolished these inhibitory effects. Consistently, RUNX2, BGLAP and SPP1 mRNA and protein expression were decreased in agomiR-103 treated

**Data Availability Statement:** All relevant data are within the manuscript and its Supporting Information files.

**Funding:** The authors received no specific funding for this work.

**Competing interests:** The authors have declared that no competing interests exist.

**Abbreviations:** HBMScs, human bone marrow mesenchymal stem cells; miRNAs, microRNAs; SATB2, SATB homeobox 2; CCK, 8cell counting Kit-8; RUNX2, RUNX family transcription factor 2; ALP, alkaline phosphatase; BGLAP, bone gamma-carboxyglutamate protein; SPP1, secreted phosphoprotein 1; UTRs, untranslated regions.

HBMScs compared with those in agomiR-NC group. Meanwhile, antagomiR-103 upregulated the mRNA and protein expression levels of RUNX2, BGLAP and SPP1 in HBMScs. Further studies revealed that *SATB2* was a direct target gene of miR-103. BMSCs transfected with agomiR-103 exhibited significantly downregulated protein expression level of SATB2, whereas knockdown of miR-103 promoted it. Additionally, rescue assays confirmed that silencing of *SATB2* partially reversed the effects of antagomiR-103 induced HBMScs proliferation and osteogenic differentiation.

## Conclusions

The present results suggested that miR-103 negatively regulates *SATB2* to serve an inhibitory role in the proliferation and osteogenic differentiation of HBMScs, which sheds light upon a potential therapeutic target for treating bone-related diseases.

## Introduction

microRNAs (miRNAs) are a family of highly conserved, endogenously expressed, single-stranded small non-coding RNAs of ~22 nucleotides in length [1]. miRNAs act as critical regulators of gene expression through binding with the 3'-untranslated regions (UTRs) of associated mRNAs [2]. Numerous previous studies demonstrated that miRNAs are involved in a variety of cellular biological behaviors, such as proliferation, migration, autophagy, and differentiation, and aberrant expression of miRNAs can lead to the development of multiple human diseases [3]. Notably, recent studies have indicated that miRNAs are related to the dysfunction of bone metabolism [4]. Osteoporosis is a progressive systemic skeletal disease in aged people characterized by low mineral density and microarchitecture deterioration of bone tissues [5]. According to previous epidemiological statistics, osteoporosis significantly increases the risk of fractures, and causes a serious social burden worldwide [6]. Osteogenic differentiation has been regarded as a critical issue in fracture therapy of osteoporosis; however, its mechanisms remain largely unclear.

Stem cells therapy provides a promising novel approach for repairing defective tissues or organs through the transplantation of cells. Dental mesenchymal stem cells have gained considerable attention as an attractive source for maxillofacial regenerative therapy [7,8]. Human bone marrow mesenchymal stem cells (HBMScs) are pluripotent stem cells that possess multiple differentiation potential, including chondrocytes, osteoblasts, fibroblasts, and adipocytes. Increasing data suggested that the HBMScs osteogenic differentiation is modulated by hormones, medicines, as well as growth factors [9]. Of note, several recent studies have shown that abnormal expression of miRNAs is relevant to HBMScs osteogenic differentiation [10]. For instance, Li et al [11] found that miR-21 facilitates osteogenesis of mouse BMScs by rgulating Smad7-Smad1/5/8-Runx2 pathway. Zhang et al [12] showed that miR-664a-5p promotes osteogenic differentiation of HBMScs by directly downregulating high-mobility group A2 expression. Xiao et al [13] reported that miR-483-3p regulates osteogenic differentiation of mouse BMScs by regulation of signal transducer and activator of transcription 1. Additionally, Li et al [14] demonstrated that miR-92b-5p modulates melatonin-mediated osteogenic differentiation of mouse BMScs by targeting intracellular adhesion molecule-1. Therefore, identification of the potential mechanisms underlying osteogenic differentiation of HBMScs is a meaningful process that developed novel therapeutic targets for osteoporosis treatment.

miR-103 is one of the members of the miR-15/107 family [15]. Several previous reports indicated that miR-103 is involved in various human diseases, including malignancies [16], nervous system disease [17], as well as fatty liver disease [18]. Chen et al [19] found that miR-103 expression was markedly downregulated in serum samples of osteoporotic patients. Valassi et al [20] reported that circulating miR-103 is associated with bone parameters in patients with controlled acromegaly. Additionally, evidence from microarray information showed that miR-103 is significantly disregulated in senescent BMScs [21]. SATB homeobox 2 (*SATB2*), as a member of AT-rich binding proteins family, is a special transcription factor that improved transcription by binding with nuclear matrix-attachment regions. *SATB2* was found to be a critical factor of osteoblast differentiation in bone development [22]. Nonetheless, the regulatory relationship between miR-103 and *SATB2* is still unclear. Thus, the present study aimed to explore the role of miR-103 on the proliferation and osteogenic differentiation of HBMScs using gain- and lose-of-function assays, and to investigate the underlying molecular mechanism of miR-103 on *SATB2*. The findings suggested that miR-103 inhibits HBMScs proliferation and osteogenic differentiation by directly targeting *SATB2*, indicating regulation of miR-103 as a potential molecular therapeutic strategy for osteoporosis treatment.

## Materials and methods

### Cell culture and osteogenic differentiation

HBMScs were obtained from Cyagen Biosciences (HUXMA-01001, Guangzhou, China). Cells were conventional cultured in OriCell® HBMScs Complete Medium (HUXMA-90011, Cyagen Biosciences) supplemented with 10% of fetal bovine serum, 100 U/ml penicillin and 100 µg/ml streptomycin, and maintained at 37˚C in a humidified atmosphere under 5% $CO_2$. For osteogenic differentiation, cells were treated with OriCell® Osteogenic Induction Differentiation Medium (HUXMA-90021, Cyagen Biosciences) at 37˚C and 5% $CO_2$ for 21 d incubation, and the medium was replaced every 2 d.

### Oligonucleotides synthesis and cell transfection

The agomiR-103, agomiR-NC, antagomiR-103, and antagomiR-NC were designed and synthesized by GenePharma (Shanghai, China). *SATB2* siRNA and corresponding siRNA NC were acquired from Hanbio Biotechnology Co., Ltd (Shanghai, China). All oligonucleotides were dissolved to suitable concentration in diethylpyrocarbonate-treated water. HBMScs in logarithmic growth phase were trypsinized and seeded in 6-well plates. When HBMScs grew to 60% cell confluence, cell were transfected with these oligonucleotides at final concentration of 200 nM using Lipofectamine 2000 Transfection Reagent (Thermo Fisher Scientific, Carlsbad, CA, USA) along with Opti-MEM Reduced Serum Medium (Thermo Fisher Scientific), according to the manufacturer's protocols. The cells were harvested 48 h after transfection for further experiments.

### Alkaline phosphatase (*ALP*) staining

The calcium deposition by osteoblasts was assessed using ALP staining. HBMSCs were placed in 12-well plates and induced osteogenic differentiation for 21 d. Then HBMSCs were rinsed three times with PBS for three times and fixed with 75% alcohol (Sigma, St Louis, CA, USA) for 30 minutes at 37˚C. The fixed cells were soaked in BCIP/NBT solution (Yeasen, Shanghai, China) and washed with PBS, and then were observed with an Olympus SP-500UZ Digital Camera (Nikon, Japan).

## *ALP* activity assay

21 days after osteogenic induction, HBMSCs were lysed with 100 μl of cell lysate buffer (21101ES60, Yeasen, Shanghai, China) for 5 min at 4˚C at 37˚C. Thereafter, the supernatant was added into 40 μl of the ALP substrate reaction solution (Yeasen) for 30 min at 37˚C in the dark according to the manufacturer's protocols, and 150 μl of the stop buffer (Yeasen) was added to stop the reaction. Finally, the optical density (OD) values were determined by measuring the absorbance at a wavelength of 405 nm using a microplate reader (Molecular Devices LLC, Sunnyvale, CA, USA). The total protein content ratio demonstrates the amounts of *ALP* produced by differentiated HBMSCs.

## Enzyme-linked immunosorbent assay (ELISA)

HBMSCs was collected after oligonucleotides treatment for 48 h. The SATB2 protein expression was calculated using a commercially available ELISA kit (abx383034, Abbexa, Cambridge, UK), according to the manufacturer's instructions.

## Alizarin red S staining

HBMSCs were placed in 12-well plates and induced osteogenic differentiation induced osteogenic differentiation for 21 d. Mineralization of cells was detected using alizarin red S Staining. Briefly, HBMSCs were trypsinized and collected, and fixed with 75% ethanol for 1 h at 37˚C. Then the cells were incubated using 40 mM alizarin red S (HUXMA-90021, Cyagen Biosciences) for 20 min at 37˚C. After washes with PBS twice to rinse needless unbound stains, the stained matrix was photographed with an Olympus SP-500UZ Digital Camera (Nikon). Five images were randomly selected and analyzed for quantification of staining with an Image Pro plus 6.0 software (Media Cybernetics, Rockville, MD, USA).

## Real-time qPCR

Total RNA was obtained from HBMScs using Trizol Reagent (Thermo Fisher Scientific), and reverse transcribed into first-stranded cDNA sequences using a miRNA cDNA Synthesis Kit (YB130911-25, Ybscience, Shanghai, China) or a PrimeScript™ RT reagent kit (DRR037A, TaKaRa, Japan). The amplification of miR-103, *SATB2*, RUNX family transcription factor 2 (*RUNX2*), bone gamma-carboxyglutamate protein (*BGLAP*), and secreted phosphoprotein 1 (*SPP1*) was performed by qPCR using the cDNA as a template on the ABI 7900HT Real-Time PCR System 7900 (Applied Biosystems, Carlsbad, CA, USA). The primers of the genes were listed in Table 1. Thermocycling conditions of PCR amplifications were performed in duplicate at 98˚C for 15 s, followed by 40 cycles of 95˚C for 10 s, 60˚C for 30 s, and 72˚C for 45 s, finally 4˚C for 30 min. The relative levels of target gene expression were quantified using the $2^{-\Delta\Delta Cq}$ method [23]. small nuclear U6 RNA was used as an internal reference for miR-103 expression analysis, and glyceraldehyde-3-phosphate dehydrogenase (*GAPDH*) served as an internal control for other genes.

## Western blot

HBMScs were lysed using mammalian protein extraction RIPA reagent (Yeasen). Equivalent amounts (50 μg) of cell protein lysates were electrophoresed on an 10% SDS-polyacrylamide gel and transferred to PVDF membranes (Thermo Fisher Scientific). The membranes were incubated with primary antibodies of RUNX (1:750, ab76956, Abcam, Cambridge, MA, USA), BGLAP (1:1000, ab13421, Abcam), SSP1 (1:750, ab69498, Abcam), and GAPDH (1:1000, 60004-1-Ig, ProteinTech Group, Chicago, IL, USA). After incubating with HRP-associated

**Table 1. Forward and reverse primers listed for real-time qPCR.**

| Name | Sequence (5'-3') |
|---|---|
| miR-103 forward | AGCAGCATTGTACAGGGCTATGA |
| miR-103 reverse | AAGGCGAGACGCACATTCTT |
| SATB2 forward | CCTGGCCCTGGGGTATTCT |
| SATB2 reverse | GTGCATCTGTCACATAACTGAGG |
| RUNX2 forward | TCTTAGAACAAATTCTGCCCTTT |
| RUNX2 reverse | GCTTTGGTCTTGAAATCACA |
| BGLAP forward | GCAGCTTGGTGCACACCTAG |
| BGLAP reverse | GGAGCTGCTGTGACATCCAT |
| SPP1 forward | CTTTCACTCCAATCGTCCCTA |
| SPP1 reverse | GCTCTCTTTGGAATGCTCAAGT |
| U6 forward | CTCGCTTCGGCAGCACA |
| U6 reverse | AACGCTTCACGAATTTGCGT |
| GAPDH forward | ATTTGGTCGTATTGGGCG |
| GAPDH reverse | TGGAAGATGGTGATGGGATT |

second antibodies, protein blots were observed using ECL Select Western Blotting Detection System (GE Healthcare, Buckinghamshire, UK) and the results were quantified using Image Pro Plus 6.0 (Media Cybernetics).

## Cell counting Kit-8 (CCK-8) assay

To measure *in vitro* growth of HBMScs, a CCK-8 assay was performed. A total of $7 \times 10^3$ cells were seeded into each well of 96-well plates and transfected with agomiR-103, agomiR-NC, antagomiR-103, and antagomiR-NC. On the 0, 24, 48, and 72 h of the measurement, each well was replaced with 100 µl of fresh DMEM medium containing 10 µl of the CCK-8 reagent (Dojindo Laboratories, Kumamoto, Japan). Plates were incubated at 37°C in a humidified atmosphere for 4 h, then the medium was shaken for 20 min. Absorbances were then measured with a microplate reader (Molecular Devices LLC) at a wavelength of 450 nm.

## Bioinformatics analysis and reporter vector construction

Bioinformatics analysis was conducted to predict the putative targets of miR-103 using TargetScan (http://www.targetscan.org) and miranda (http://www.microrna.org). The wild type (WT) 3'-UTR of *SATB2* contained the predicted miR-103 binding sites and a mutant (MUT) 3'-UTR of *SATB2* were obtained from Hanbio Biotechnology Co., Ltd. The two sequences were inserted into the psiCHECK™2 vector (Promega Corporation, Madison, WI, USA) to produce SATB2 WT and MUT reporter vectors, respectively.

## Luciferase reporter assay

HBMScs cells at 60% confluence in 6-well plates were co-transfected with 50 nM agomiR-103, agomiR-NC, antagomiR-103, and antagomiR-NC, along with 0.75 µg *SATB2* WT or MUT reporter vectors using Lipofectamine 2000 Transfection Reagent. The Firefly and Renilla luciferase activities were measured 48 h after transfection using a Dual-Luciferase Reporter Assay System (E1910, Promega Corporation), according the manufacturer's protocols. Renilla luciferase acted as a reporter gene and Firefly luciferase as a normalized internal reference for each individual analysis.

## Statistical analysis

SPSS 19.0 software (SPSS Inc., Chicago, IL, USA) was used for the statistical analyses. Each experiment was conducted and repeated for three times. The quantitative data were listed as the mean ± standard deviation. Independent Student's t-tests and one-way ANOVA analyses were used to compare differences between groups. Differences with $p$-values of less than 0.05 were considered to be statistically significant.

## Results

### The expression of miR-103, *RUNX2*, *BGLAP* and *SPP1* in HBMScs after induction of osteogenic differentiation

To assess the association between miR-103 and osteogenic differentiation, HBMScs were induced osteogenic differentiation for 21 d incubation, and the expression levels of miR-103 in HBMScs after 0, 7, 14, and 21 d induction were observed. Our results showed that the associated osteogenic differentiation biomarkers (*RUNX2*, *BGLAP*, and *SPP1*) mRNA expression was significantly upregulated in HBMScs on days 7, 14, and 21 as compared with undifferentiated cells on 0 day (Fig 1A–1C, $p<0.05$ or $p<0.001$), indicating osteogenic differentiation of HBMScs is successful. Notably, the results of real-time qPCR revealed that the expression level of miR-103 was markedly reduced in HBMScs during osteogenic differentiation in a time-dependent way (Fig 1D, $p<0.05$ or $p<0.001$). Consistently, the protein expression of RUNX, BGLAP and SSP1 was increased in HBMScs during osteogenic differentiation (Fig 1E and 1F, $p<0.05$ or $p<0.01$ or $p<0.001$). The data suggested that decreased miR-103 expression correlates with HBMScs osteogenic differentiation.

### The modulation of miR-103 on the proliferation of HBMScs

To determine the expression levels of miR-103 following oligonucleotides transfection, real-time qPCR was conducted. The results revealed that the expression of miR-103 was 102.55 times higher in HBMScs treated with agomiR-103 than in cells transfected with the agomiR-NC group (Fig 2A, $p<0.001$). Also, miR-103 expression was downregulated by 76.18% in HBMScs transfected with antagomiR-103 compared with the antagomiR-NC group (Fig 2B, $p<0.001$). Further, the effects of the miR-103 overexpression and knockdown on the proliferation of HBMScs were determined. The results indicated that the proliferation abilities of HBMScs decreased by 14.62%, 21.70% and 34.89% at 24, 48 and 72 h posttransfection of the agomiR-103, as compared with the agomiR-NC group (Fig 2C, $p<0.05$ or $p<0.001$). Meanwhile, in HBMScs treated with antagomiR-103 the results revealed that at 24, 48 and 72 h the cell proliferation was increased by 11.24%, 30.15% and 29.44%, compared with cells transfected with the antagomiR-NC group (Fig 2D, $p<0.05$ or $p<0.001$). These data validated that miR-103 suppresses the proliferation of HBMScs.

### The effects of miR-103 overexpression and knockdown on HBMScs osteogenic differentiation

Furthermore, to better identify the functional effects of miR-103 overexpression and knockdown on the osteogenic differentiation of HBMScs, the expressions of *RUNX2*, *BGLAP*, and *SPP1* were detected. As presented in Fig 3A, the mRNA expression levels of RUNX2, BGLAP, and SPP1 were markedly reduced during osteogenic differentiation in the agomiR-103 group compared with the agomiR-NC group ($p<0.05$ or $p<0.001$). Compared with the antagomiR-NC group, these genes expression were significantly increased in the antagomiR-103 treated HBMScs (Fig 3B, $p<0.001$). Western blot showed that agomiR-103 treatment

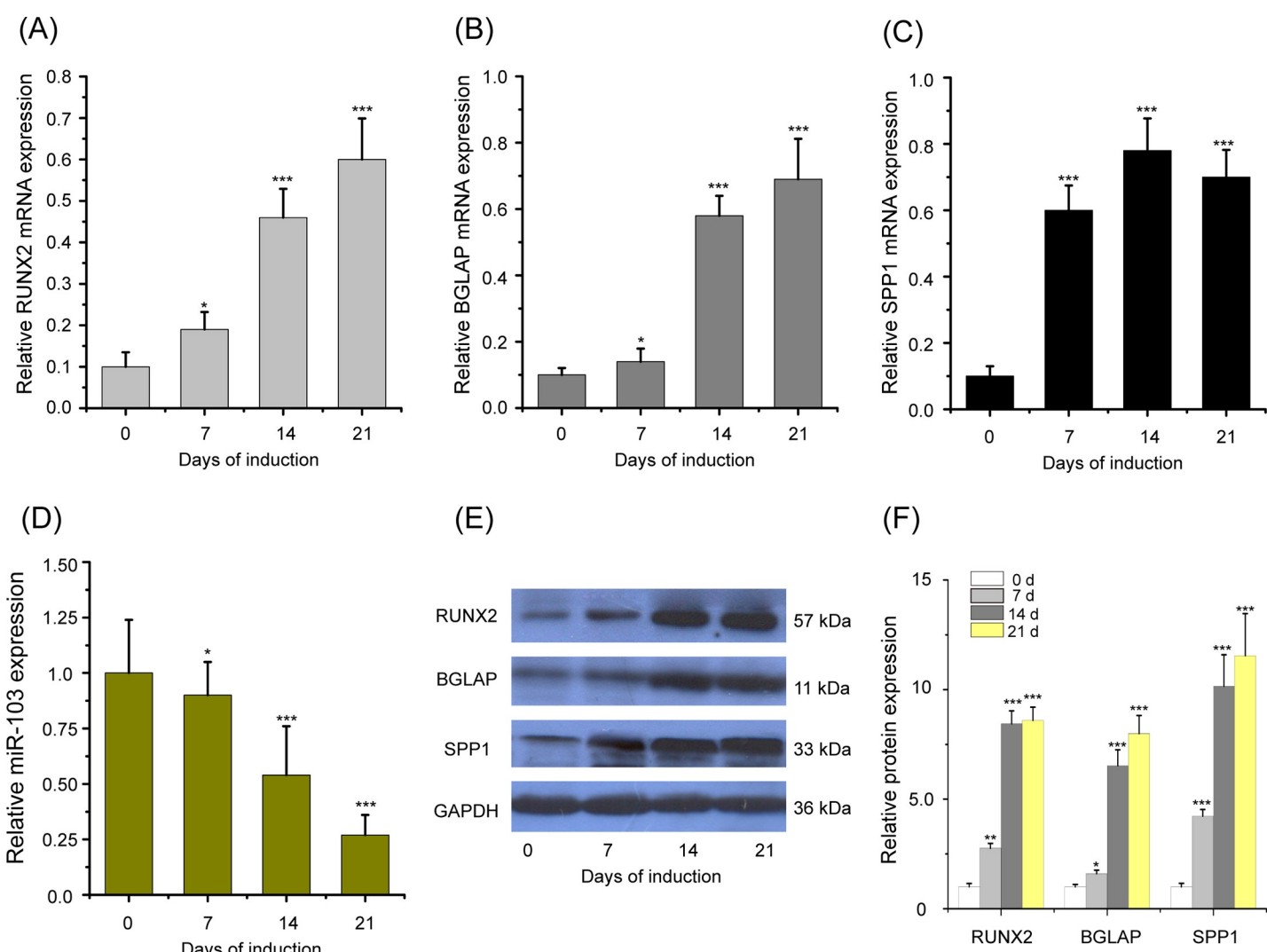

**Fig 1. The expression of miR-103, *RUNX2*, *BGLAP* and *SPP1* in HBMScs after induction of osteogenic differentiation.** Real-time qPCR was performed to detect the mRNA expression levels of *RUNX2* (A), *BGLAP* (B) and *SPP1* (C) in HBMScs after 0, 7, 14, and 21 d induction. (D) The expression of miR-103 was markedly reduced in HBMScs during osteogenic differentiation in a time-dependent way. (E and F) Western blot analysis of the protein expression of RUNX, BGLAP and SSP1 in HBMScs during osteogenic differentiation. Columns mean of three independent experiments, and bars SD. *$p<0.05$, **$p<0.01$, ***$p<0.001$.

decreased RUNX2, BGLAP, and SPP1 protein expression, whereas antagomiR-103 induced these proteins expression (Fig 3C and 3D, $p<0.05$ or $p<0.001$). To corroborate the roles of miR-103 on the osteogenic process, *ALP* activity and alizarin red S staining were performed. As expected, overexpression of miR-103 decreased *ALP* activity and mineralized-bone matrix formation in HBMScs (Fig 3E and 3F, $p<0.001$). Consistently, silencing of miR-103 exhibited higher *ALP* activity and mineralized-bone matrix formation after osteogenic differentiation of 21 days (Fig 3G and 3H, $p<0.001$). Thus, the above data confirmed that miR-103 negatively regulates the osteogenic differentiation of HBMScs.

## The negatively regulation of miR-103 on the potential target gene *SATB2*

*SATB2* mRNA 3'-UTR has a potential complimentary binding sites to miR-203a (Fig 4A). Real-time qPCR and ELISA were applied to investigate the relationship between the expression

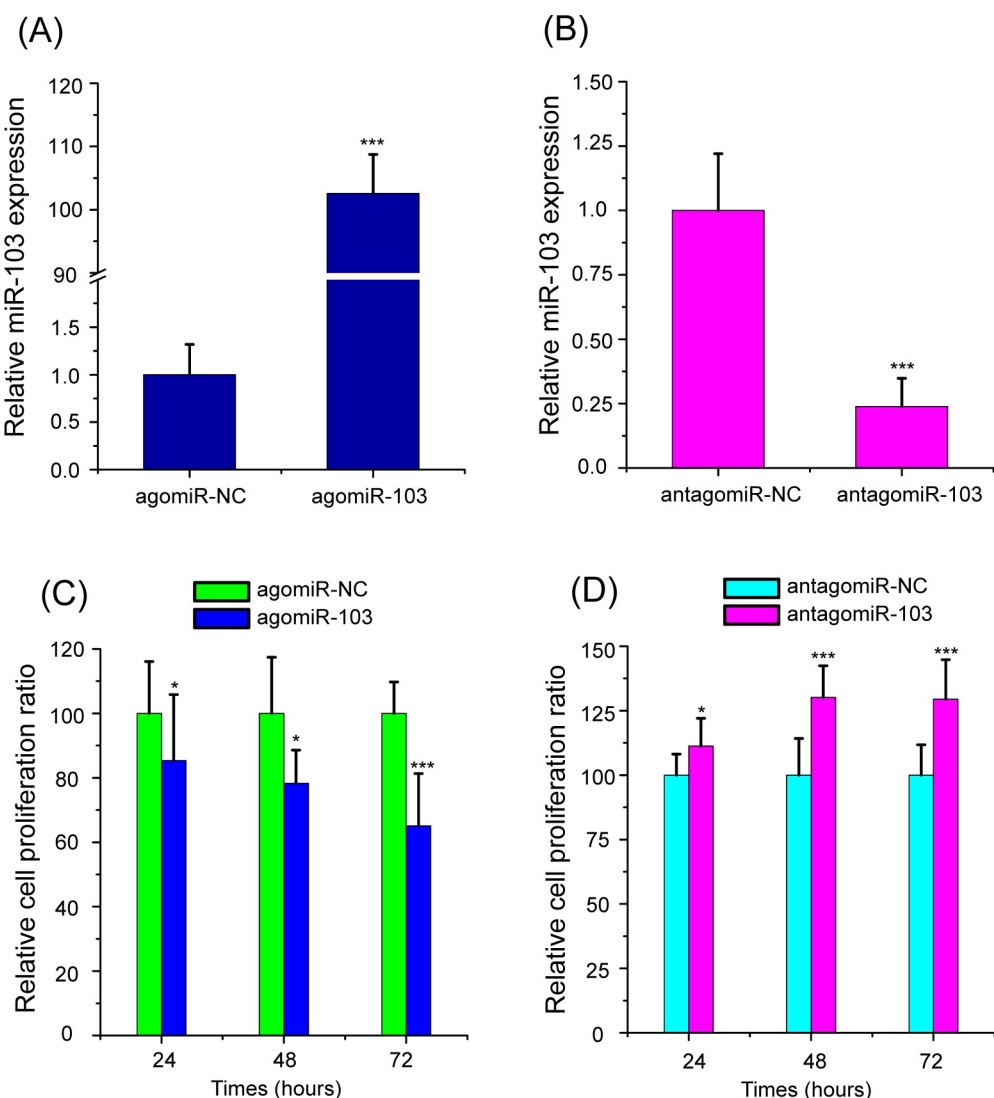

**Fig 2. miR-103 suppresses the proliferation of HBMScs.** (A) Elevated expression of miR-103 was determined by real-time qPCR when HBMScs were transfected with agomiR-103. (B) The expression of miR-103 was significantly downregulated in BMMSCs after treated with antagomiR-103. (C) A CCK-8 assay demonstrated that overexpression of miR-103 retarded the growth of HBMScs *in vitro*. (D) Knockdown of miR-103 promoted HBMScs proliferation ability. Data are shown as the mean ± SD of three independent experiments. *$p<0.05$, ***$p<0.001$.

levels of miR-103 and SATB2. As shown in Fig 4B and 4C, enforced miR-103 expression led to downregulate the mRNA and protein expression levels of SATB2 in HBMSCs; by contrast, application of antagomiR-103 to the HBMScs markedly upregulated SATB2 mRNA and protein expression ($p<0.01$ or $p<0.001$). Additional, a dual luciferase reporter assay was performed to validated the miR-103 binding sites on the 3'-UTR of *SATB2* mRNA, and WT and MUT *SATB2* reporter plasmids were constructed according to this sites. Results showed that in WT *SATB2* reporter plasmid the relative luciferase activity significantly decreased in agomiR-103-transfected HBMScs (Fig 4D, $p<0.001$). With MUT *SATB2* reporter plasmid, there was no significant difference in relative luciferase activity between the agomiR-103 and agomiR-NC groups. Conversely, miR-103 knockdown substantially escalated the relative luciferase activity of the *SATB2* reporter plasmid that carried the WT but not MUT 3'-UTR of

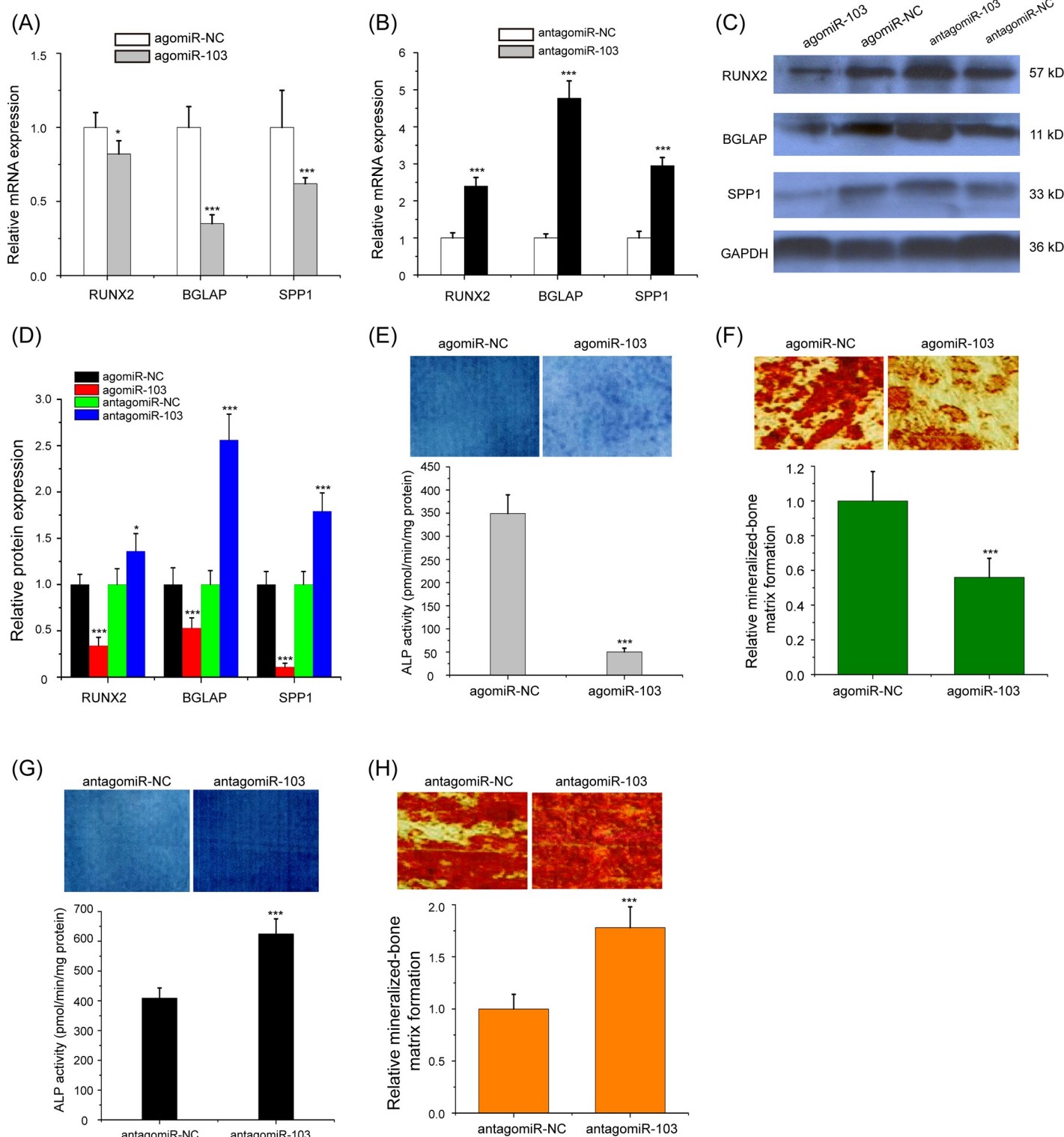

**Fig 3. miR-103 represses the osteogenic differentiation of HBMScs.** (A) Real-time qPCR analysis detected mRNA expression of *RUNX2*, *BGLAP* and *SPP1* in HBMScs after treatment with agomiR-103 or agomiR-NC. (B) The *RUNX2*, *BGLAP* and *SPP1* mRNA expression were significantly increased in the antagomiR-103 treated HBMScs. (C and D) Western blot analysis of the protein expression of RUNX, BGLAP and SSP1 in HBMScs by miR-103 overexpression and knockdown. (E) The ALP activity of HBMScs was significantly suppressed when miR-103 was overexpressed. (F) The osteogenic differentiation of HBMScs transfected with agomiR-103 and agomiR-NC was observed by Alizarin red S staining. (G) Knockdown of miR-103 exhibited higher ALP activity after osteogenic differentiation of 21 days. (H) Alizarin red S staining investigated the ability of mineralized-bone matrix formation in HBMScs treated with antagomiR-103 or antagomiR-NC. Columns mean of three independent experiments. $^{*}p<0.05$, $^{***}p<0.001$.

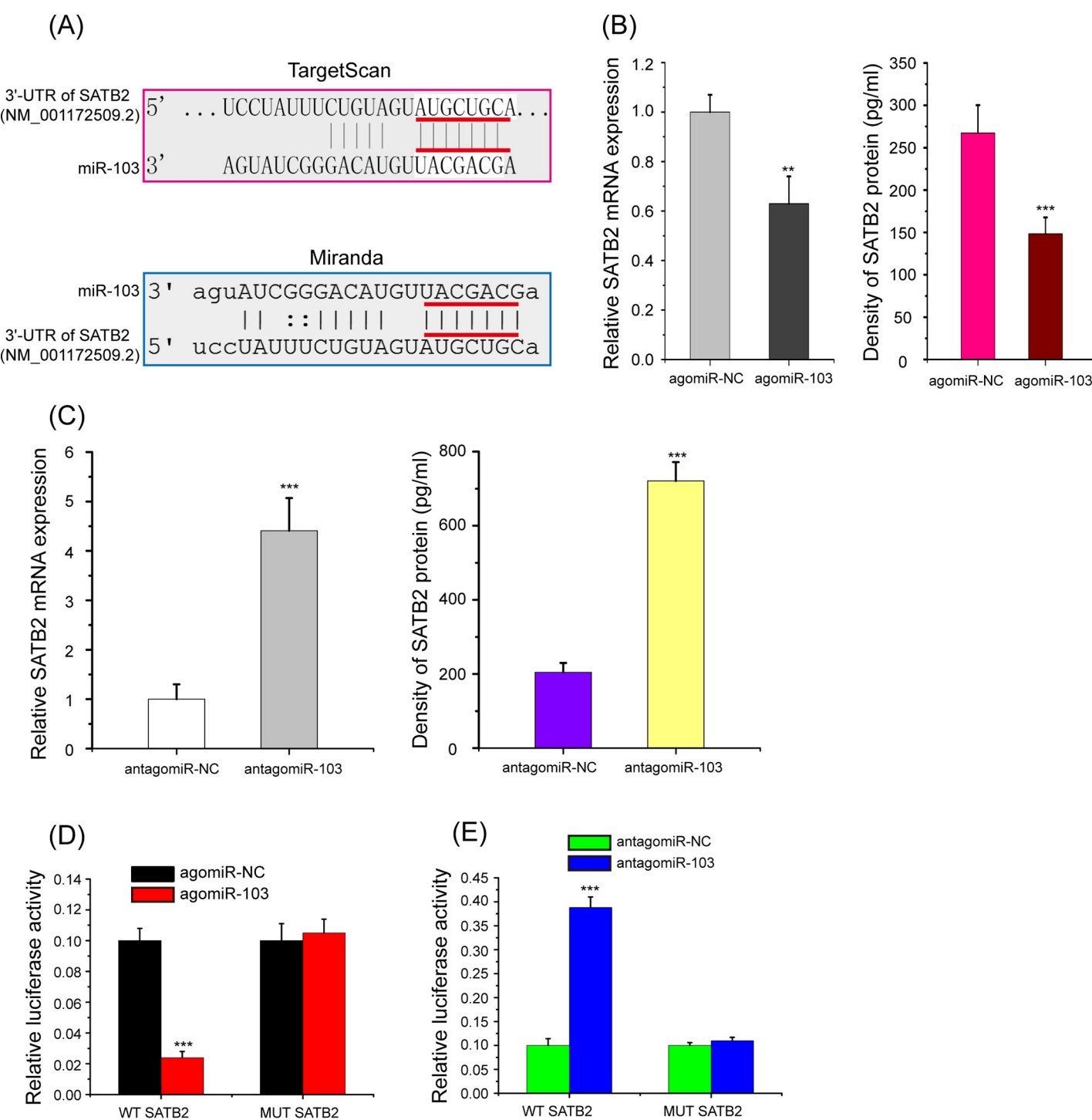

**Fig 4. SATB2 is a direct target of miR-103 in HBMScs.** (A) miR-103 target sites in the 3'-UTR of *SATB2* mRNA. (B) Real-time qPCR and ELISA were conducted to examine the effects of miR-103 overexpression on the mRNA and protein expression of SATB2. GAPDH was also detected as a loading control. (C) Knockdown of miR-103 markedly upregulated SATB2 mRNA and protein expression in HBMScs. (D) Relative luciferase activity of the WT or MUT *SATB2* reporter plasmids along with agomiR-103 or agomiR-NC in HBMScs was shown. Renilla luciferase activity was normalized to Firefly luciferase activity and plotted as relative luciferase activity. (E) Luciferase activities were measured in HBMScs following cotransfection with the WT or MUT *SATB2* reporter plasmids and antagomiR-103 or antagomiR-NC. Data are shown as the mean ± SD of three independent experiments. $^{**}p<0.01$, $^{***}p<0.001$.

*SATB2* (Fig 4E, p<0.001). These findings meant that *SATB2* is a direct target of miR-103 in HBMScs.

## The regulatory mechanism of miR-103 on HBMScs proliferation and osteogenic differentiation by directly downregulating *SATB2*

As *SATB2* plays a critical role in osteoblast differentiation of bone development, and *SATB2* is a direct target of miR-103, we speculated that miR-103 may perform its functions by regulating *SATB2*. To verify the participation of *SATB2* in the effects of miR-103 on the proliferation and osteogenic differentiation, HBMScs with miR-103 knockdown were transfected with either the *SATB2* siRNA or corresponding siRNA NC, and SATB2 expression was detected by real-time qPCR and ELISA. The data of results confirmed that the protein and mRNA expression levels of SATB2 in HBMScs were significantly decreased following SATB2 siRNA transfection, and SATB2 siRNA partly reversed the upregulation of SATB2 expression by antagomiR-103(Fig 5A and 5B, p<0.01 or p<0.001). Interesting, HBMScs co-transfected with antagomiR-103 and SATB2 siRNA exhibited a lower cell proliferation ability than cells transfected with antago-miR-103 and siRNA NC (Fig 5C, p<0.001). Moreover, antagomiR-103 induced HBMScs osteogenic differentiation could be partly abolished by *SATB2* knockdown, as evidenced by reduction in *ALP* activity and mineralized matrix formation (Fig 5D and 5E, p<0.05 or p<0.001), and reduced protein expression levels of RUNX2, BGLAP, and SPP1 (Fig 5F and 5G, p<0.01 or p<0.001). Altogether, these data demonstrated miR-103 participates in the proliferation and osteogenic differentiation of HBMScs by directly targeting *SATB2*.

## Discussion

BMScs are a commonly used progenitor cell source to study osteogenic differentiation in progressive systemic skeletal diseases because of their capacity for self-renewal and differentiation potential. During the past years, a number of miRNAs have been determined as key regulators of osteogenic differentiation of BMScs by modulating a series of genes expression involved in bone development [24]. Recently, study by Wang and his colleagues [25] found that steroid-induced avascular necrosis of the femoral head exhibits reduced osteogenic differentiation and promotes fat differentiation, and elevates miR-103 expression. Shen and his team members [26] reported that miR-103 expression is consistently downregulated in the forkhead transcription factor C1-overexpressing MC3T3-E1 cells. Given these findings, we explored the potential function and mechanism of miR-103 in regulating proliferation and osteogenic differentiation. We selected HBMScs to perform our experiments, and our *in vitro* results supported that miR-103 negatively regulates *SATB2* to serve an inhibitory role in the proliferation and osteogenic differentiation of HBMScs. Our findings provides more insights into the function and mechanism of miRNAs in modulating the osteogenesis of HBMScs.

miR-103 is a notable miRNA that it is evolutionarily conserved and involved in regulating cell differentiation, metabolism and immune [27]. For example, Croizier et al [28] demonstrated that miR-103/107 signal controls developmental switch of proopiomelanocortin progenitors into neuropeptide Y neurons and impacts glucose homeostasis. In another study, Holik and his team [29] reported that n($\epsilon$)-carboxymethyllysine promotes the miR-103/143 expression and enhances lipid accumulation in 3T3-L1 cells. In addition, recent study reported that shenmai injection improves the postoperative immune function by inhibiting differentiation into Treg cells via miR-103/GPER1 axis [30]. However, researches concerning miR-103 on osteogenesis are still very limited. Here, we examined the effect and mechanism of miR-103 on HBMScs osteogenic differentiation because we detected decreased levels of miR-103 in a time-dependent way after induction of osteogenic differentiation. Similar to our findings, Yoo

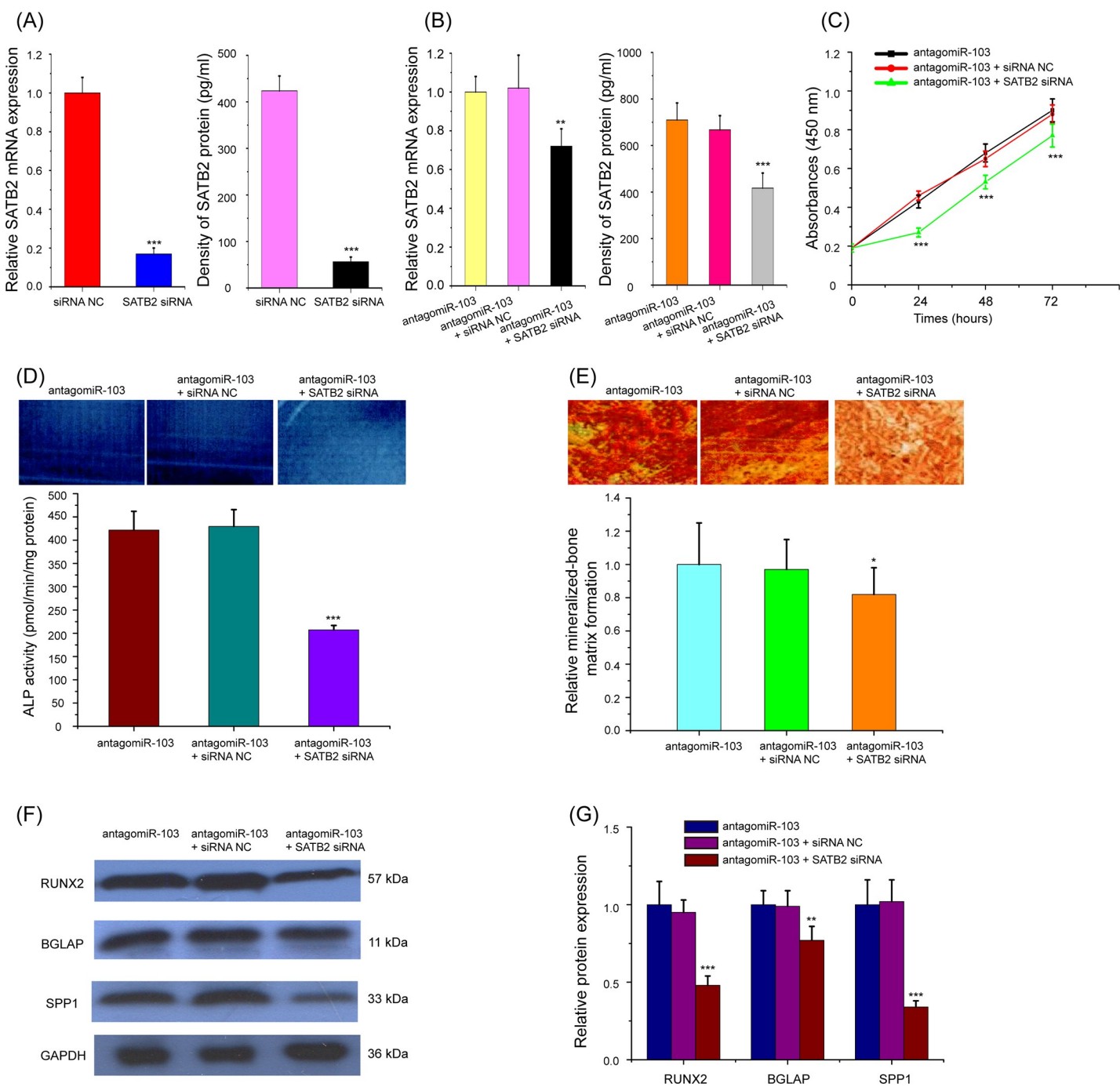

**Fig 5. miR-103 participates in the proliferation and osteogenic differentiation of HBMScs by directly targeting *SATB2*.** (A) Expression levels of SATB2 mRNA and protein were examined after transfection of *SATB2* siRNA and siRNA NC. (B) Transfection of *SATB2* siRNA partly reversed the upregulation of SATB2 expression by antagomiR-103. (C) HBMScs co-transfected with antagomiR-103 and *SATB2* siRNA exhibited a lower cell proliferation ability than cells transfected with antagomiR-103 and siRNA NC. (D) *SATB2* knockdown partly abolished antagomiR-103 induced *ALP* activity of HBMScs. (E) *SATB2* knockdown partly abolished antagomiR-103 induced mineralized matrix formation of HBMScs. (F and G) HBMScs treatment with antagomiR-103 and SATB2 siRNA reduced the protein expression levels of RUNX2, BGLAP, and SPP1 compared cells treated with antagomiR-103 and siRNA NC. Columns mean of three independent experiments. $^{*}p<0.05$, $^{**}p<0.01$, $^{***}p<0.001$.

et al [21] recently reported expression profiles of miRNAs in senescent BMScs. It has previously been demonstrated that upregulation expression of some osteogenic markers, including *RUNX2*, *BGLAP*, and *SPP1*, and accumulated mineralization of the extracellular matrix can be observed during the osteogenic differentiation [31]. Similarly, our study revealed that overexpression of miR-103 led to the decrease of *ALP* activity and mineralized-bone matrix formation, and reduction of RUNX2, BGLAP, and SPP1 expression, and conversely, while silencing of miR-103 elevated these phenomenon. Additional, we found HBMScs proliferation was arrested by miR-103, which is in agreement with previous study in the cancer field, where miR-103 has been shown to functions as a tumor suppressor gene in non-small cell lung cancer by directly targeting programmed cell death 10 [32]. These data confirmed that miR-103 negatively regulates the proliferation and osteogenic differentiation of HBMScs.

To further to decipher mechanism, computational bioinformatics analysis was taken to identify target mRNAs that that could explain the phenotypic effects inhibited by miR-103 on HBMScs proliferation and osteogenic differentiation. One target gene of interest was the *SATB2* gene (Gene ID: 23314), which located at 2q33.1 and have 18 exons. *SATB2* is a member of the highly conserved AT-rich binding proteins family, and its protein is involved in transcription regulation and chromatin remodeling [33]. The mutation or deficiency of *SATB2* in humans is associated with several severe diseases, such as craniofacial deformity, craniosynostosis, and mental retardation [34]. Indeed, *SATB2* is documented to be a sensitive biomarker for colorectal carcinoma [35], pancreatic cancer [36] and osteosarcoma [37]. Recently, accumulating evidences confirmed that *SATB2* serves a key role in bone development and osteoblastic differentiation [22]. Notably, *SATB2* has been found to be targeted by several miRNAs, including miR-31 [38], miR-34a/b/c [39,40], miR-383 [41], and miR-875-5p [42]. Here, by luciferase reporter and ELISA assay, we identified that miR-103 directly bound to the *SATB2* mRNA 3'-UTR and suppressed SATB2 expression in HBMScs. Moreover, rescue assays confirmed that silencing of *SATB2* partially rescued the effects of miR-103 knockdown induced HBMScs proliferation and osteogenic differentiation. Our study is not without limitations. We could not determine miR-103 expression and the correlation between miR-103 and *SATB2* in osteoporosis patients. Further studies are warranted to identify the regulatory mechanism of miR-103 on *SATB2* in inhibiting HBMScs proliferation and osteogenic differentiation *in vivo*.

Taken together, the present study demonstrated that miR-103 suppresses the proliferation and osteogenic differentiation of HBMScs by directly targeting *SATB2*. Thus, targeting miR-103 may be a molecular therapeutic strategy for bone-related diseases treatment.

## Supporting information

**S1 Fig. Western blot analysis of the protein expression of RUNX, BGLAP and SSP1 in HBMScs.**
(TIF)

## Author Contributions

**Conceptualization:** Yuanrui Wang.

**Formal analysis:** Hao Lv, Huashan Yang.

**Methodology:** Hao Lv, Huashan Yang, Yuanrui Wang.

**Supervision:** Hao Lv.

**Writing – original draft:** Hao Lv.

**Writing – review & editing:** Yuanrui Wang.

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
