## [Decision Letter · Decision Letter 0]

17 Jan 2020

PONE-D-19-35932

Effects of miR-103 by negatively regulating SATB2 on proliferation and osteogenic differentiation of human bone marrow mesenchymal stem cells

PLOS ONE

Dear Mr. Wang,

Thank you for submitting your manuscript to PLOS ONE. After careful consideration, we feel that it has merit but does not fully meet PLOS ONE’s publication criteria as it currently stands. Therefore, we invite you to submit a revised version of the manuscript that addresses the points raised during the review process.

This manuscript is of interest but it presents some critical point that need to be addressed in full. The Authors therefor in their revision must highlight all the changes made without any exception and answer to each point raised.

We would appreciate receiving your revised manuscript by Mar 02 2020 11:59PM. To enhance the reproducibility of your results, we recommend that if applicable you deposit your laboratory protocols in protocols.io, where a protocol can be assigned its own identifier (DOI) such that it can be cited independently in the future. For instructions see: http://journals.plos.org/plosone/s/submission-guidelines#loc-laboratory-protocols

We look forward to receiving your revised manuscript.

Kind regards,

Gianpaolo Papaccio, M.D., Ph.D.

Academic Editor

PLOS ONE

Journal Requirements:

2. Please include a copy of Table 1 which you refer to in your text on page 10.

Additional Editor Comments (if provided):

Reviewers' comments:

Reviewer's Responses to Questions

**Comments to the Author**

1. Is the manuscript technically sound, and do the data support the conclusions?

Reviewer #1: Partly

Reviewer #2: Yes

2. Has the statistical analysis been performed appropriately and rigorously? 

Reviewer #1: I Don't Know

Reviewer #2: Yes

3. Have the authors made all data underlying the findings in their manuscript fully available?

Reviewer #1: Yes

Reviewer #2: Yes

4. Is the manuscript presented in an intelligible fashion and written in standard English?

Reviewer #1: Yes

Reviewer #2: Yes

5. Review Comments to the Author

Reviewer #1: In this manuscript the Authors investigate about the role on miR-103 in the osteogenic differentiation. The Authors propose that miR-103 interact and promote the degradation of SATB2 mRNA. Blocking miR-103 with a specific antagomir promote bone differentiation by increasing RUNX, BGLAP and SSP1, ALP activity, and matrix formation. The work is interesting in principle, however there are few concerns that need to be addressed as follows:

Fig.1 show the inverse correlation between miR-103 and RUNX, BGLAP and SSP1 expression, this data should be validated at protein level by WB, IF etc. The axis report “mRMA relative expression” I believe the Authors meant mRNA, please correct.

Fig3 A,B shows that treatment with antagomiR-103 and agomiR-103 influence RUNX, BGLAP and SSP1 expression, this result should be validated at protein level by WB,IF etc. same mistake as above about “mRMA”

Fig.3 C,E report the effect of antagomiR-103 and agomiR-103 on ALP activity, image of the cells should be shown together with plots (e.g. colorimetric assay).

Fig 3 D,F report the effect of antagomiR-103 and agomiR-103 on matrix mineralization, image of alizarin red should be shown together with plots.

Fig 5 show the effect of siRNA and anatgomiR-103 on SATB2, all this result must be validated at protein level by WB,IF etc.

Without the proper validation al these results are not reliable.

Reviewer #2: In this paper authors examined the effect and mechanism of miR-103 on HBMScs osteogenic differentiation and demonstrated the relationship between miR103 and SATB2 in osteogenic differentiation.

The paper is interesting but some corrections are needed.

Authors should add information about the therapeutic potential resulting from the use of stem cells in osteogenic differentiation (Cells. 2019 Mar; 8(3): 217; Stem Cells Dev. 2019 Aug 1;28(15):1050-1058)

Authors should indicate how many days they evaluated the effects of miR-103 overexpression and knockdown on HBMScs osteogenic differentiation through q-RT PCR.

Authors should present not only quantification but also images of Alizarin red.

Authors should perform also protein expression of osteogenic genes, such as BGLAP, OPN, BSP, after overexpression and knockdown of miR-103, through WB or IHC/IF staining.

6. PLOS authors have the option to publish the peer review history of their article (what does this mean?). If published, this will include your full peer review and any attached files.

Reviewer #1: No

Reviewer #2: No

---

## [Author Response · Author response to Decision Letter 0]

1 Apr 2020

Dear Editors and Reviewers: 

Thank you for your letter and for the reviewers’ comments concerning our manuscript. Those comments are all valuable and very helpful for revising and improving our paper. We have studied comments carefully and have made correction which we hope meet with approval. The entire review has undergone major changes. The main corrections in the paper highlighted in blue. The responds to the reviewer’s comments are as flowing:

Sincerely Yours,

Yuanrui Wang, Department of Trauma Center, Jinan Central Hospital Affiliated to Shandong First Medical University

E-mail: wangyrjinan@sina.com.

Journal Requirements:

Reply: We have revised the manuscript style as required.

2. Please include a copy of Table 1 which you refer to in your text on page 10.

Reply: We have added Table 1 in the revised manuscript as required.

Review Comments to the Author

Reviewer #1: In this manuscript the Authors investigate about the role on miR-103 in the osteogenic differentiation. The Authors propose that miR-103 interact and promote the degradation of SATB2 mRNA. Blocking miR-103 with a specific antagomir promote bone differentiation by increasing RUNX, BGLAP and SSP1, ALP activity, and matrix formation. The work is interesting in principle, however there are few concerns that need to be addressed as follows:

Fig.1 show the inverse correlation between miR-103 and RUNX, BGLAP and SSP1 expression, this data should be validated at protein level by WB, IF etc. The axis report “mRMA relative expression” I believe the Authors meant mRNA, please correct.

Reply: We have performed WB experiment to detect the protein expression of RUNX, BGLAP and SSP1 in HBMScs after induction of osteogenic differentiation. Consistently, the protein expression of RUNX, BGLAP and SSP1 was increased in HBMScs during osteogenic differentiation (Fig 1E and 1F, p<0.05 or p<0.01 or p<0.001). The “mRMA” has been corrected to “mRNA” in all Figures.

Fig3 A,B shows that treatment with antagomiR-103 and agomiR-103 influence RUNX, BGLAP and SSP1 expression, this result should be validated at protein level by WB,IF etc. same mistake as above about “mRMA”

Reply: We have performed WB experiments to detect the protein expression of RUNX, BGLAP and SSP1 in HBMScs after treatment with antagomiR-103 and agomiR-103. Consistent with mRNA expression characteristics, we found that overexpression of miR-103 decreased the protein expression of RUNX, BGLAP and SSP1, while miR-103 knockdown increased these proteins expression.

Fig.3 C,E report the effect of antagomiR-103 and agomiR-103 on ALP activity, image of the cells should be shown together with plots (e.g. colorimetric assay).

Reply: We have added the images of alkaline phosphatase staining in the revised Fig 3.

Fig 3 D,F report the effect of antagomiR-103 and agomiR-103 on matrix mineralization, image of alizarin red should be shown together with plots.

Reply: We have added the images of alizarin red staining in the revised Fig 3.

Fig 5 show the effect of siRNA and anatgomiR-103 on SATB2, all this result must be validated at protein level by WB,IF etc. Without the proper validation all these results are not reliable.

Reply: The mRNA and protein expression of SATB2 in HBMScs after transfection of SATB2 siRNA and anatgomiR-103 had been validated by real-time qPCR and ELISA assays. We have performed WB experiments to detect the protein expression of RUNX, BGLAP and SSP1 after transfection of SATB2 siRNA and anatgomiR-103. we found that SATB2 siRNA partly reversed the upregulation of RUNX, BGLAP and SSP1 protein expression by antagomiR-103.

Reviewer #2: In this paper authors examined the effect and mechanism of miR-103 on HBMScs osteogenic differentiation and demonstrated the relationship between miR103 and SATB2 in osteogenic differentiation.

The paper is interesting but some corrections are needed.

Authors should add information about the therapeutic potential resulting from the use of stem cells in osteogenic differentiation (Cells. 2019 Mar; 8(3): 217; Stem Cells Dev. 2019 Aug 1;28(15):1050-1058)

Reply: We have added these information and references in the revised manuscript (page 5-6, lines 110-113).

Authors should indicate how many days they evaluated the effects of miR-103 overexpression and knockdown on HBMScs osteogenic differentiation through q-RT PCR.

Reply: HBMSCs were induced osteogenic differentiation for 21 d.

Authors should present not only quantification but also images of Alizarin red.

Reply: We have added the images of alizarin red staining in the revised Fig 3.

Authors should perform also protein expression of osteogenic genes, such as BGLAP, OPN, BSP, after overexpression and knockdown of miR-103, through WB or IHC/IF staining.

Reply: We have performed WB experiments to detect the protein expression of RUNX, BGLAP and SSP1 in HBMScs after treatment with antagomiR-103 and agomiR-103. Consistent with mRNA expression characteristics, we found that overexpression of miR-103 decreased the protein expression of RUNX, BGLAP and SSP1, while miR-103 knockdown increased these proteins expression.

---

## [Decision Letter · Decision Letter 1]

21 Apr 2020

Effects of miR-103 by negatively regulating SATB2 on proliferation and osteogenic differentiation of human bone marrow mesenchymal stem cells

PONE-D-19-35932R1

Dear Dr. Wang,

We are pleased to inform you that your manuscript has been judged scientifically suitable for publication and will be formally accepted for publication once it complies with all outstanding technical requirements.

With kind regards,

Gianpaolo Papaccio, M.D., Ph.D.

Academic Editor

PLOS ONE

Additional Editor Comments (optional):

Reviewers' comments:

Reviewer's Responses to Questions

**Comments to the Author**

1. If the authors have adequately addressed your comments raised in a previous round of review and you feel that this manuscript is now acceptable for publication, you may indicate that here to bypass the “Comments to the Author” section, enter your conflict of interest statement in the “Confidential to Editor” section, and submit your "Accept" recommendation.

Reviewer #1: All comments have been addressed

Reviewer #2: All comments have been addressed

2. Is the manuscript technically sound, and do the data support the conclusions?

Reviewer #1: Yes

Reviewer #2: (No Response)

3. Has the statistical analysis been performed appropriately and rigorously? 

Reviewer #1: I Don't Know

Reviewer #2: (No Response)

4. Have the authors made all data underlying the findings in their manuscript fully available?

Reviewer #1: Yes

Reviewer #2: (No Response)

5. Is the manuscript presented in an intelligible fashion and written in standard English?

Reviewer #1: Yes

Reviewer #2: (No Response)

6. Review Comments to the Author

Reviewer #1: the authors addressed all the concerns raised by this reviewer, now the manuscript is greatly improved.

Reviewer #2: (No Response)

7. PLOS authors have the option to publish the peer review history of their article (what does this mean?). If published, this will include your full peer review and any attached files.

Reviewer #1: No

Reviewer #2: No

---

## [Editor Report · Acceptance letter]

27 Apr 2020

PONE-D-19-35932R1 

Effects of miR-103 by negatively regulating SATB2 on proliferation and osteogenic differentiation of human bone marrow mesenchymal stem cells 

Dear Dr. Wang:

I am pleased to inform you that your manuscript has been deemed suitable for publication in PLOS ONE. Congratulations! Your manuscript is now with our production department. 

With kind regards,

on behalf of

Prof. Gianpaolo Papaccio 

Academic Editor

PLOS ONE